# IMPROVING SEARCH THROUGH A3C REINFORCEMENT LEARNING BASED CONVERSATIONAL AGENT

## ABSTRACT

We develop a reinforcement learning based search assistant which can assist users through a set of actions and sequence of interactions to enable them realize their intent. Our approach caters to subjective search where the user is seeking digital assets such as images which is fundamentally different from the tasks which have objective and limited search modalities. Labeled conversational data is generally not available in such search tasks and training the agent through human interactions can be time consuming. We propose a stochastic virtual user which impersonates a real user and can be used to sample user behavior efficiently to train the agent which accelerates the bootstrapping of the agent. We develop A3C algorithm based context preserving architecture which enables the agent to provide contextual assistance to the user. We compare the A3C agent with Q-learning and evaluate its performance on average rewards and state values it obtains with the virtual user in validation episodes. Our experiments show that the agent learns to achieve higher rewards and better states.

## 1 INTRODUCTION

Within the domain of search, the recent advances have focused on personalizing the search results through recommendations (Wei et al., 2017; Shani et al., 2005). While the quality of recommendations have improved, the conventional search interface has not innovated much to incorporate useful contextual cues which are often missed. Conventional search interface enables the end user to perform a keyword based faceted search where the typical work flow goes as follows: the end user types in her search query, applies some filters and then modifies the query based on the results. This iterative interaction naturally paves way for incorporating conversations in the process. Instead of the search engine just retrieving the best result set, it can interact with the user to collect more contextual cues. For example, if a user searches for birthday gift, the search engine could follow-up by asking who are you buying the gift for. Such information and interaction can provide more human-like and engaging search experience along with assisting user in discovering their search intent. In this work we address this problem by developing a Reinforcement Learning (RL) (Sutton & Barto, 1998) based conversational search agent which interacts with the users to help them in narrowing down to relevant search results by providing them contextual assistance.

RL based dialogue agents have been designed for tasks like restaurant, bus and hotel reservation (Peng et al., 2017) which have limited and well-defined objective search modalities without much scope for subjective discussion. For instance, when searching for a restaurant, the user can specify her preferences (budget, distance, cuisines etc) due to which the problem can be modeled as a slot filling exercise. In contrast, suppose a designer is searching for digital assets (over a repository of images, videos etc) to be used in a movie poster. She would start with a broad idea and her idea would get refined as the search progresses. The modified search intent involves an implicit cognitive feedback which can be used to improve the search results. We model our agent for this type of search task. Since the user preferences can not be modeled using a fixed set of facets, we end up with a very large search space which is not the case with most other goal oriented RL agents.

We model the search process as a sequence of alternate interactions between the user and the RL agent. The extent to which the RL agent could help the user depends on the sequence and the type of actions it takes according to user behavior. Under the RL framework, intermediate rewards is given to the agent at each step based on its actions and state of conversational search. It learns

the applicability of different actions through these rewards. In addition to extrinsic rewards, we define auxiliary tasks and provide additional rewards based on agent's performance on these tasks. Corresponding to the action taken by the agent at each turn, a response is selected and provided to the user.

Since true conversational data is not easily available in search domain, we propose to use query and session log data to develop a stochastic virtual user environment to simulate training episodes and bootstrap the learning of the agent. Our agent interacts with the user to gauge user intent and treats the search engine as a black box service which makes it easily deployable over any search engine. We perform qualitative experiments by simulating validation episodes with different reinforcement learning algorithms under various formulations of the state space to evaluate the performance of the trained agent.

Our contributions are three-fold: 1) formulating conversational interactive search as a reinforcement learning problem and proposing a generic and easily extendable set of states, actions and rewards; 2) developing a stochastic user model which can be used to efficiently sample user actions while simulating an episode; 3) we develop A3C (Asynchronous Advantage Actor-Critic) (Mnih et al., 2016) algorithm based architecture to predict the policy and state value functions of RL agent and compare it with other RL algorithms over performance on validation episodes.

## 2 RELATED WORK

There have been various attempts at modeling conversational agents, as dialogue systems (Cuayáhuitl, 2017; Zhao & Eskenazi, 2016; Ultes et al., 2017; Levin et al., 1997) and text-based chat bots (Cuayáhuitl & Dethlefs, 2011; Li et al., 2016a;b; Vinyals & Le, 2015; Weston et al., 2014). Some of these have focused on modeling goal driven RL agent such as indoor way finding system (Cuayáhuitl & Dethlefs, 2011) to assist humans to navigate to their destination and visual input agents which learn to navigate and search an object in a 3-D environment space (Zhu et al., 2017).

Domain independent RL based dialogue systems have been explored in the past. For example, (Ultes et al., 2017) uses User Satisfaction (US) as the sole criteria to reward the learning agent and completely disregards Task Success(TS). But US is a subjective metric and is much harder to measure or annotate real data with. In our formulation, we provide a reward for task success at the end of search along with extrinsic and auxiliary rewards at intermediate steps (discussed in section 3.4). Other RL based information seeking agents extract information from the environment by sequentially asserting questions but these have not been designed on search tasks involving human interaction and behavior(Bachman et al., 2016).

Neural Conversation Model, based on the SEQ2SEQ framework, uses an end-to-end and domain independent encoder-decoder architecture to maximize the likelihood of next utterance given the previous utterances (Vinyals & Le, 2015). The resulting model generates grammatically correct sentences though they tend to be repetitive, less engaging and lack consistency such that it cannot perform coherent and meaningful conversation with real humans. To overcome these issues, deep RL has been combined with Neural Conversation Model to foster sustained conversations based on the long-term success of dialogues (Li et al., 2016b). The model is initialized with MLE parameters and tuned further using policy gradient (Sutton et al., 2000) through rewards which capture ease of answering, information flow and semantic coherence. Neural conversation models initialized with MLE parameters have also been improved using batch policy gradient method which efficiently uses labeled data comprising of scores assigned to each dialogue in the conversation (Kandasamy et al., 2017). These models require labeled conversational data which is not available for the subjective search tasks we discussed. Our model assists the user at different steps in the search through a set of assist actions instead.

RL has also been used for improving document retrieval through query reformulation where the agent sequentially reformulates a given complex query provided by the user (Nogueira & Cho, 2017; Narasimhan et al., 2016). But their work focuses on single turn episodes where the model augments the given query by adding new keywords. In contrast, our agent engages the user directly into the search which comprises of sequence of alternate turns between user and agent with more degrees of freedom (in terms of different actions the agent can take).

To minimize human intervention while providing input for training such agents in spoken dialogue systems, simulated speech outputs have been used to bypass spoken language unit (Cuayáhuitl, 2017). The system uses word based features obtained from raw text dialogues to represent the state of the Reinforcement Learning Agent. This approach enables to reduce the system's dependence on hand engineered features. User models for simulating user responses have been obtained by using LSTM which learns inter-turn dependency between the user actions. They take as input multiple user dialogue contexts and outputs dialogue acts taking into account history of previous dialogue acts and dependence on the domain (Asri et al., 2016).

Another prominent research area, closely related to conversational agents, is that of question answering. In case of machine comprehension, the task is to reason over multiple statements to formulate the answer to a given query. *Memory Networks* Weston et al. (2014) use memories to store information (facts and contexts) internally in a compressed form for later use. RL has been used in visual tasks (Strub et al., 2017; Das et al., 2016) where given an image, history of the utterances and a follow-up question about the image, the agent is trained to answer the question. Unlike machine comprehension based question answering tasks, Visual Dialog tasks are closer to conversational agents as they require the model to maintain and employ the context of the conversation when formulating the replies. But unlike Visual Dialog, our responses are not grounded in the given image and require knowledge beyond the immediate context of the conversation.

Often task oriented dialogue systems are difficult to train due to absence of real conversations and subjectivity involved in measuring shortcomings and success of a dialogue Dodge et al. (2015). Evaluation becomes much more complex for subjective search systems due to absence of any label which tells whether the intended task had been completed or not. We evaluate our system through rewards obtained while interacting with the user model and also on various real world metrics (discussed in experiments section) through human evaluation.

## 3 SYSTEM MODEL

In this paper, we experiment with two different RL algorithms - the Asynchronous Advantage Actor Critic (A3C) algorithm (Mnih et al., 2016) and the Q-learning (Whitehead, 2014). We first discuss preliminaries of RL, then provide the details of the action-state spaces, rewards and virtual user we modeled followed by discussion of the above algorithms and our architecture.

### 3.1 REINFORCEMENT LEARNING

Reinforcement Learning is the paradigm to train an agent to operate in an environment **E**. The agent interacts with the environment in a series of independent episodes and each episode comprises of a sequence of turns. At each turn, the agent observes the state $s$ of the environment ($s \in \mathbf{S}$, where **S** is defined as the *state space* - the set of possible states) and performs an action $a$ ($a \in \mathbf{A}$, where **A** is defined as the action space - the set of all the possible actions). When the agent performs an action, the state of the environment changes and the agent gets the corresponding reward (Sutton & Barto, 1998). An optimal policy maximizes cumulative reward that the agent gets based on the actions taken according to the policy from start till the final terminal state is reached in the episode.

### 3.2 AGENT ACTION SPACE

Action space **A** is designed to enable the search agent to interact with the user and help her in searching the desired assets conveniently. The agent actions can be divided into two sets - the set of probe intent actions - **P** and general actions - **G** as described in Table 1 and Table 2 respectively. The agent uses the probe intent actions **P** to explicitly query the user to learn more about her context. For instance, the user may make a very open-ended query resulting in a diverse set of results even though none of them is a good match. In such scenarios, the agent may prompt the user to refine her query or add some other details like where the search results would be used. Alternatively, the agent may cluster the search results and prompt the user to choose from the clustered categories. These actions serve two purposes - they carry the conversation further and they provide various cues about the search context which is not evident from the search query provided by the user.

Table 1: Probe intent actions

| Action | Description |
|---|---|
| probe use case | ask about where assets will be used |
| probe to refine | ask the user to further refine query if less relevant search results are retrieved |
| cluster categories | ask the user to select from categorical options related to her query |

Table 2: General actions

| Action | Description |
|---|---|
| show results | display results corresponding to most recent user query |
| add to cart | suggest user to bookmark assets for later reference |
| ask to download | suggest user to download some results if they suit her requirement |
| ask to purchase | advise the user to buy some paid assets |
| provide discount | offer special discounts to the user based on search history |
| sign up | ask the user to create an account to receive updates regarding her search |
| ask for feedback | take feedback about the search so far |
| provide help | list possible ways in which the agent can assist the user |
| salutation | greet the user at the beginning; say goodbye when user concludes the search |

The set **G** consists of generic actions like displaying assets retrieved corresponding to the user query, providing help to the user etc. While probe intent actions are useful to gauge user intent, set **G** comprises of actions for carrying out the functionality which the conventional search interface provides like "presenting search results". We also include actions which promote the business use cases (such as prompting the user to signup with her email, purchase assets etc). The agent is rewarded appropriately for such prompts depending on the subsequent user actions. Our experiments show that the agent learns to perform different actions at appropriate time steps in search episodes.

### 3.3 STATE SPACE

We model the state representation in order to encapsulate facets of both search and conversation. The state $s$ at every turn in the conversation is modeled using the history of user actions - $history\_user$,[1] history of agent actions - $history\_agent$, discretized relevance scores of search results - $score\_results$ and a variable $length\_conv$ which represents number of user responses in the conversation till that point.

The variables $history\_user$ and $history\_agent$ comprises of user and agent actions in last $k$ turns of the conversational search respectively. This enables us to capture the context of the conversation (in terms of sequence of actions taken). Each user-action is represented as one-hot vector of length 9 (which is the number of unique user actions). Similarly, each agent-action has been represented as a one-hot vector of length 12. The history of the last 10 user and agent actions is represented as concatenation of these one-hot vectors. We use vectors with zero padding wherever needed such as when current history comprises of less than 10 turns.

The variable $score\_results$ quantifies the degree of similarity between most recent query and the top 10 most relevant search assets retrieved. They have been taken in state representation to incorporate the dependency between the relevance of probe intent actions and quality of search results retrieved. Similarly, $length\_conv$ has been included since appropriateness of other agent actions like $sign\ up$ may depend on the duration for which the user has been searching.

---

[1] $history\_user$ includes most recent user action to which agent response is pending in addition to remaining history of user actions.

### 3.4 REWARDS

Reinforcement Learning is concerned with training an agent in order to maximize some notion of cumulative reward. In general, the action taken at time $t$ involves a long term versus short term reward trade-off leading to the classic exploration-exploitation problem. This problem manifests itself even more severely in the context of conversational search. For instance, let us say that the user searches for "nature". Since the user explicitly searched for something, it would seem logical that the most optimal action is to provide the search results to the user. Alternatively, instead of going for immediate reward, the agent could further ask the user if she is looking for "posters" or "portraits" which would help in narrowing down the search in the long run. Determining the most optimal action at any point of the conversation is a non-trivial task which highlights the importance of reward modeling.

Since we aim to optimize dialogue strategy and do not generate dialogue utterances, we assign the rewards corresponding to the appropriateness of the action performed by the agent considering the state and history of the search. We have used some rewards such as task success (based on implicit and explicit feedback from the user during the search) which is also used in PARADISE framework (Walker et al., 1997). At the same time several metrics used by the PARADISE cannot be used for evaluating our system or modeling rewards. For instance, time required (number of turns) for user to search desired results cannot be penalized since it can be possible that user is finding the system engaging and helpful in refining the results better which may increase number of turns in the search. We model the total reward which the agent gets in one complete dialogue comprises of three kinds of rewards and can be expressed in the form of following equation :

$$R_{total}(search) = r_{TaskCompletion}(dialogue) + \sum_{t \in turns} \left( r_{extrinsic}(t) + r_{auxiliary}(t) \right)$$

#### 3.4.1 TASK COMPLETION AND EXTRINSIC REWARDS

First kind of reward ($r_{TC}$) is based on the completion of the task (Task Completion TC) which is download and purchase in the case of our search problem. This reward is provided once at the end of the episode depending on whether the task is completed or not. As second kind of rewards, we provide instantaneous extrinsic rewards (Deci et al., 1999) - ($r_{extrinsic}$) based on the response that the user gives subsequent to an agent action. Rewards provided on the basis of interaction with simulated user have been studied and compared with inverse RL previously (Chandramohan et al., 2011). We categorize the user action into three feedback categories, namely good, average or bad. For example, if the agent prompts the user to refine the query and the user does follow the prompt, the agent gets a high reward because the user played along with the agent while if the user refuses, a low reward is given to the agent. A moderate reward will be given if the user herself refines the query without the agent's prompt. Depending on these feedback categories, $r_{extrinsic}$ is provided at every time step in the search.

#### 3.4.2 AUXILIARY REWARDS

Apart from the extrinsic rewards, we define a set of auxiliary tasks $T_A$ specific to the search problem which can be used to provide additional reward signals, $r_{auxiliary}$, using the environment. We define $T_A = \{$# click result, # add to cart, # cluster category click, if sign up option exercised$\}$. $r_{auxiliary}$ is determined and provided at every turn in the search based on the values of different auxiliary tasks metrics defined in $T_A$ till that turn in the search. Such rewards promotes a policy which improves the performance on these tasks.

### 3.5 TRAINING RL AGENT THROUGH STOCHASTIC USER MODEL

The RL agent is trained to learn the optimal action policy which requires actual conversational search data which is not available as conversational agents have not been used in the context of search task we defined. To bypass this issue and bootstrap training, we propose a user model that simulates user behavior to interact with the agent during training and validation. Our methodology can be used to model a virtual user using any query and log sessions data.

We developed a stochastic environment where the modeled virtual human user responds to agent's actions. The virtual human user has been modeled using query sessions data from a major stock

| Session Data | Mapped User Action |
|---|---|
| shopping ; content_type : all ; NO_OFFSET ; search | *new query* |
| shopping ; content_type : all ; 100 ; search | *request more* |
| child while shopping ; content_type:all ; NO_OFFSET ; search | *refine query* |
| child while shopping ; content_type:all ; NO_OFFSET ; click | *click result* |
| child while shopping ; content_type: landscape ; NO_OFFSET ; search | *cluster category click* |

Figure 1: Example of mapping session data to user actions. The session data comprises of sequence of logs, each log comprises of search query, filters applied (content type), offset field and interaction performed by the user (such as search, click etc)

photography and digital asset marketplace which contain information on queries made by real users, the corresponding clicks and other interactions with the assets. This information has been used to generate a user which simulates human behavior while searching and converses with the agent during search episode. We map every record in the query log to one of the user actions as depicted in Table 3. Figure 1 shows an example mapping from session data to user action. To model our virtual user, we used the query and session log data of approximately 20 days.

Table 3: Mapping between query logs and user actions

| User action | Mapping used |
|---|---|
| new query | first query or most recent query with no intersection with previous ones |
| refine query | query searched by user has some intersection with previous queries |
| request more | clicking on next set of results for same query |
| click result | user clicking on search results being shown |
| add to cart | when user adds some of searched assets to her cart for later reference |
| cluster category click | when user clicks on filter options like orientation or size |
| search similar | search assets with similar series, model etc |

The virtual user is modeled as a finite state machine by extracting conditional probabilities - $P(User\ Action\ u|\ History\ h\ of\ User\ Actions)$. These probabilities are employed for sampling next user action given the fixed length history of her actions in an episode. The agent performs an action in response to the sampled user action. Subsequent to the action performed by the agent, next user action is sampled which modifies the state and is used to determine the reward the agent gets for its previous action. Table 4 shows a snippet of conditional probability matrix of user actions given the history of last 3 user actions.

Table 4: Snippet of conditional probability matrix obtained from session data on query logs

| User action | User action history | P(user action / history) |
|---|---|---|
| Click assets | More assets, click assets, similar click | 0.41 |
| More assets | New query, refine query, add to cart | 0.13 |
| Refine query | similar click, new query, new query | 0.40 |

The query and session log data has been taken from an asset search platform where the marketer can define certain offers/promotions which kick in when the user takes certain actions, for instance the user can be prompted to add some images to cart (via a pop-up box). Users response to such prompts on the search interface is used as proxy to model the effect of RL agent on virtual user's sampled action subsequent to different probe actions by the agent. This ensures that our conditional

probability distribution covers the whole probability space of user behavior. In order to incorporate the effect of other agent actions such as sign up which are not present in the query logs, we tweaked the probability distribution realistically in order to bootstrap the agent.

## 3.6 Q-LEARNING

The agent can be trained through *Q-learning* (Watkins, 1989) which consists of a real valued function $Q : S \times A \to \mathbb{R}$. This Q-function maps every state-action pair $(s, a)$ to a Q-value which is a numerical measure of the expected cumulative reward the agent gets by performing $a$ in state $s$. Suppose the agent takes an action $a$ in state $s$ such that the environment makes a transition to a state $s'$, the Q-value for the pair $(s, a)$ is updated as follows:

$$Q_{i+1}(s, a) = (1 - \alpha)Q_i(s, a) + \alpha(r + \gamma max_{a'} Q_i(s', a'))$$

where $\alpha$ is the learning rate, r is the immediate reward for performing action $a$ in state $s$ in $i^{th}$ user turn in the episode. For our case, an episode refers to one complete session of conversational search between the user and the agent. Once the Q-values are learned, given a state, the action with the maximum Q-value is chosen. In order to prevent the agent from always exploiting the best action in a given state, we employ an $\epsilon-$ greedy exploration policy (Wunder et al., 2010), $0 < \epsilon < 1$. The size of our state space is of the order of $\approx 10^7$. For Q-learning, we use the table storage method where the Q-values for each state is stored in a lookup table which is updated at every step in a training episode.

## 3.7 A3C ALGORITHM

In this algorithm, we maintain a value function $V_\pi$ and a stochastic policy $\pi$ as a function of the state. The policy $\pi : A \times S \to \mathbb{R}$ defines a probability distribution $\pi(a|s)$ over the set of actions which the agent may take in state $s$ and is used to sample agent action given the state. The value function $V_\pi : S \to \mathbb{R}$ represents the expected cumulative reward from current time step in an episode if policy $\pi$ is followed after observing state s i.e. $V_\pi(s) = \mathbb{E}_{a \sim \pi(.|s)}[Q_\pi(s, a)]$.

### 3.7.1 SEARCH CONTEXT PRESERVING A3C ARCHITECTURE

We propose a neural architecture (figure 2) which preserves the context of the conversational search for approximating the policy and value functions. The architecture comprises of a LSTM (Hochreiter & Schmidhuber, 1997) which processes the state at a time step t (input $\mathbf{i_t} = \mathbf{s_t}$) and generates an embedding $\mathbf{h_t}$ which is further processed through a fully connected layer to predict the probability distribution over different actions using softmax function (Bridle, 1990) and the value of the input state. Following equations describes our architecture.

$$\mathbf{h_t} = \mathbf{f}(\mathbf{w_{LSTM}}, \mathbf{s_t})$$
$$\mathbf{o_{p_t}} = \mathbf{f}(\theta_\mathbf{p}, \mathbf{h_t})$$
$$\mathbf{p_t} = softmax(\mathbf{o_{p_t}})$$
$$v_{s_t} = f(\theta_\mathbf{v}, \mathbf{h_t})$$

, where $\mathbf{w_{LSTM}}$ represents the parameters of the LSTM, $\theta_p$ and $\theta_v$ represents the set of parameters of the last fully connected layer which outputs the policy $\mathbf{p}$ and value $v_{s_t}$ of the input state $\mathbf{s_t}$ respectively. We represent all the parameters by $\theta = \{\mathbf{w_{LSTM}}, \theta_\mathbf{p}, \theta_\mathbf{v}\}$. The LSTM state is reset to zero vectors at the start of a search episode. At time-step $t$ in the search episode, the state $s_t$ is given as input to the model. The cell and hidden state $(c_{t-1}, h_{t-1})$ of the LSTM is maintained based on the previous states $(s_0, s_1, ..., s_{t-1})$ which have already been processed. The LSTM unit remembers the previous states which enables our model to capture the effect of observed states in the search while predicting the probability of different actions. This "memory" implicitly allows our agent to make the next prediction based on the transitions and user behavior observed so far allowing it to mimic the strategy of a real agent assisting the user.

The parameters are tuned by optimizing the loss function $loss_{total}$ which can be decomposed into three types of losses.

$$loss_{total}(\theta) = loss_{policy}(\theta) + loss_{value}(\theta) + loss_{entropy}(\theta)$$

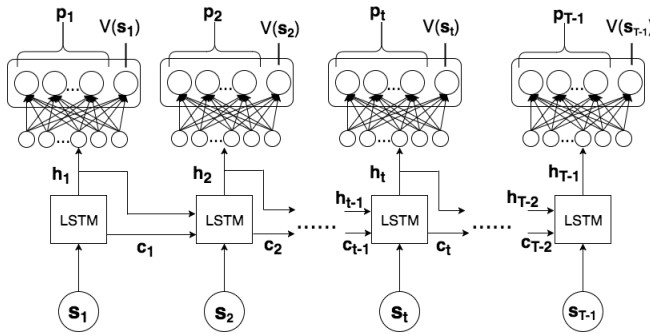

Figure 2: A3C architecture for predicting policy $\mathbf{p_t}$ and value $V(\mathbf{s_t})$. Current search state $\mathbf{s_t}$ is processed by a LSTM followed by a fully connected layer. The cell state $\mathbf{c_t}$ and hidden state $\mathbf{h_t}$ of LSTM from previous time step is retained while processing the next state during an episode. The same fully connected layer is used for prediction at different time steps in an episode. The episode terminates at time step $T$.

We now explain all the three losses. In A3C algorithm, the agent is allowed to interact with the environment to roll-out an episode. The network parameters are updated after completion of every n-steps in the roll-out. An n-step roll-out when the current state is $s_t$ can be expressed as $(s_t, a_t, r_t, s_{t+1}, v_{s_t}) \rightarrow (s_{t+1}, a_{t+1}, r_{t+1}, s_{t+1}, v_{s_{t+1}}) \rightarrow ... \rightarrow (s_{t+n-1}, a_{t+n-1}, r_{t+n-1}, s_{t+n}, v_{s_{t+n-1}})$. We also calculate $V(s_{t+n}; \theta)$ in order to estimate $loss_{value}$ which is defined as:

$$loss_{value}(\theta) = (V_{target}(s_i) - V(s_i; \theta))^2, \quad \text{for } i = t, t+1, ..., t+n-1$$
$$\text{where, } V_{target}(s_i) = \sum_{k=0}^{t+n-i-1} \gamma^k r_{k+i} + \gamma^{n+t-i} V(s_{t+n}; \theta)$$

Thus an n-step roll-out allows us to estimate the target value of a given state using the actual rewards realized and value of the last state observed at the end of the roll-out. Value of a terminal state $s_T$ is defined as 0. Each roll-out yields n samples to train the network on the value loss function using these estimated values. In a similar way, the network is trained on $loss_{policy}$ which is defined as :

$$loss_{policy}(\theta) = -\log(p(a_i | s_i; \theta)) * A(a_i, s_i; \theta), \quad \text{for } i = t, t+1, ..., t+n-1$$
$$\text{where, } A(a_i, s_i; \theta) = \sum_{k=0}^{t+n-i-1} \gamma^k r_{k+i} + \gamma^{n+t-i} V(s_{t+n}; \theta) - V(s_i; \theta)$$

The above loss function tunes the parameter in order to shift the policy in favor of actions which provides better advantage $A(a_t, s_t, \theta)$ given the state $s_t$. This advantage can be interpreted as additional reward the agent gets by taking action $a_t$ in state $s_t$ over the average value of the state $V(s_t; \theta)$ as the reference. However, this may bias the agent towards a particular or few actions due to which the agent may not explore other actions in a given state. To prevent this, we add entropy loss to the total loss function which aims at maximizing the entropy of probability distribution over actions in a state.

$$loss_{entropy}(\theta) = -\sum_{a \in \mathbf{A}} -p(a | s_i; \theta) \log(p(a | s_i; \theta)), \quad \text{for } i = t, t+1, ..., t+n-1$$

The total loss function $loss_{total}$ incorporates exploitation-exploration balance through policy and entropy loss functions optimization. The value function $V_\pi(s)$ is used for determining value of a state to be used as reference while determining advantage of different actions in $loss_{policy}$. We use Adam optimizer (Kingma & Ba, 2014) for optimizing the loss function on model parameters $\theta$. In order to improve the exploration capacity of the final agent trained, A3C comprises of a global model and uses multiple asynchronous agents which interact with their own copy of environment in parallel. Each agent uses its local gradients of the loss function with respect to model parameters to update the parameters of the global model and then copies the parameters of the global model for subsequent training. This is repeated after completion of every fixed number of episodes for each agent which results in faster convergence.

### 3.7.2 CAPTURING SEARCH CONTEXT AT LOCAL AND GLOBAL LEVEL

Including the vectors which encode the history of agent and user actions in last $k$ turns of the search to the state captures the *local context*. User behavior at current time-step can be affected by queries far away in the history. Since the search episode may extend indefinitely, local context is not sufficient to capture this behavior. The LSTM unit in our architecture aggregates the local context as it sequentially processes the states in an episode into a *global context* which results in capturing context at a global search level.

## 4 EXPERIMENTS

In this section, we evaluate the trained agent with the virtual user model and discuss the results obtained with the two reinforcement learning techniques, A3C and Q-learning, and compare them. For each algorithm, we simulate validation episodes after each training episode and plot the average rewards and mean value of the states obtained during the validation episodes. We also developed a chat-search interface (implementation details in section 6.1 in the appendix) where real users can interact with the trained agent during their search.[2] Some conversations between the trained agent and real humans using this interface have been provided in section 6.2 in the appendix.

### 4.1 A3C USING USER MODEL

The global model is obtained using 10 local agents which are trained in parallel threads (each trained over 350 episodes). We compare the validation results using this global model for different state representations for conversational search and hyper-parameter settings such as discount factor ($\gamma$) (which affects exploration vs exploitation trade-off) and the LSTM size which controls the context preserving capacity of our architecture.

### 4.1.1 VARYING DISCOUNT FACTOR

We experiment with 3 values of discount factor and fix the LSTM size to 250. Figure 3 shows the validation trend in average rewards corresponding to 3 discount factors. Greater discount factor (lower value of $\gamma$) results in lowers weights for the future rewards. With a large discount factor, the agent tries to maximize the immediate rewards by taking the greedy actions since future rewards are discounted to a larger extent. We validate this by computing the variance in the results for each case. We do not consider the values before 100 episodes as the network is under-fitting in that region. The variance values for the 3 cases ($\gamma = 0.90, 0.70, 0.60$) are $1.5267, 1.627$, and $1.725$ respectively. Since the agent takes more greedy actions with higher discount factors, the variance in the reward values also increases since the greedy approach yields good rewards in some episodes and bad rewards in others.

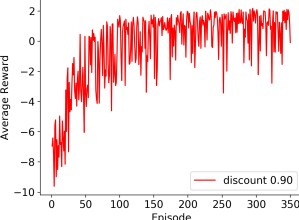 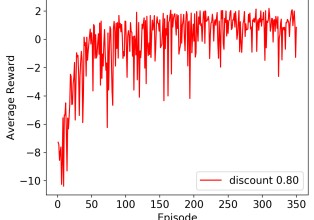 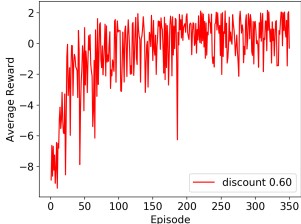

Figure 3: Plot of average validation reward against number of training episodes for A3C agent. The size of LSTM is 250 for each plot with varying discount factor; $\gamma = 0.90$ (left), $\gamma = 0.80$ (middle) and $\gamma = 0.60$ (right). It can be observed that a lower $\gamma$ value results in higher variance in the rewards resulting in a greedy (less exploratory) policy.

For further experiments, we fix the discount value to $0.90$ as this value achieves better exploration-exploitation balance.

---

[2]Supplementary material containing snapshots and demo video of the chat-search interface can be accessed at https://drive.google.com/open?id=0BzPI8zwXMOiWNk5hRElRNG4tNjQ

### 4.1.2 VARYING MEMORY CAPACITY

In the next setup, we vary the size of the LSTM as 100, 150 and 250 to determine the effect of size of the context preserved. Figure 4 depicts the trend in mean value of states observed in an episode. We observe that larger size of the LSTM results in better states which the agent observes on an average since average state value is higher. This demonstrates that a bigger LSTM size providing better capacity to remember the context results in agent performing actions which yield improved states in search.

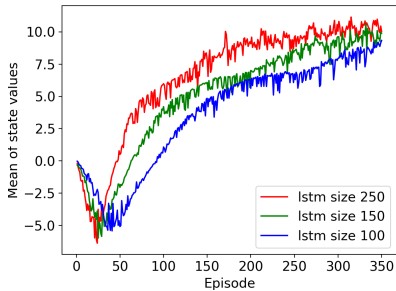

Figure 4: Plot of mean of state values observed in an episode for A3C agent. Different curves correspond to different LSTM size. The discount value is $\gamma = 0.90$ for each curve. Better states (higher average state values) are observed with larger LSTM size since it enables the agent to remember more context while performing actions.

### 4.1.3 DIFFERENT STATE REPRESENTATIONS

In this experiment, we model the state vector without incorporating the action history vectors - $history\_user$ and $history\_agent$. In Figure 5, we plot the mean state values observed in an episode and compare the case where the history of actions is added to state vector with the case where it is not added. The 3 plots correspond to different LSTM sizes. For large LSTM size (= 250), the history need not be explicitly added to the state as the LSTM is able to preserve the context and eventually achieves the same mean state values. But if the LSTM does not have enough capacity, as in case of LSTM size = 100, the mean state values observed with history vector included in the state is more than when it is not included. This demonstrates that including the local context in state representation is useful to enable the architecture to aggregate it into global context.

### 4.2 Q-LEARNING USING USER MODEL

Figure 6 shows the trend in average reward in validation episode versus number of training episodes. We experimented with values of different hyper-parameters for Q-learning such as discount ($\gamma$) and exploration control parameter ($\epsilon$) determined their optimal values to be 0.70 and 0.90 respectively based on trends in the validation curve and average reward value at convergence. We compare the A3C agent (with LSTM size 250 and $\gamma = 0.90$ (left plot in figure 3)) with the Q-learning agent (figure 6). It can be observed that the A3C agent is able to obtain better averaged awards ($\approx 2.0$) in validation episodes upon convergence as compared to the Q-agent which obtains $\approx 0.50$.

### 4.3 HUMAN EVALUATION OF AGENT TRAINED THROUGH A3C

Since A3C algorithm performs and generalize better than Q-learning approach, we evaluated our system trained using A3C algorithm in the rebuttal period through professional designers who regularly use image search site for their design tasks.

To evaluate the effectiveness of our system when interacting with professional designers, we asked them to search images which they will use while designing a poster on natural scenery using both our conversational search agent and conventional search interface provided by stock photography

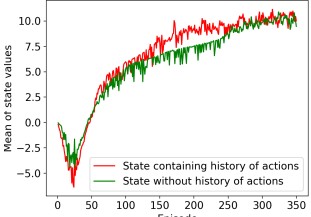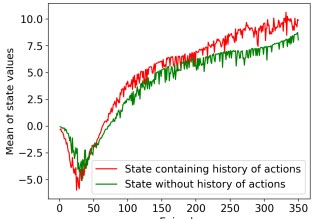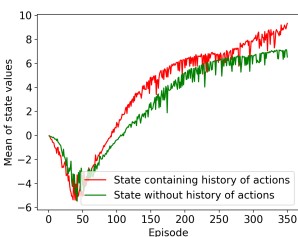

Figure 5: Plots of mean of state values in an episode for A3C agent with $\gamma = 0.90$ to analyze advantage of history in state representation. The 3 plots correspond to different LSTM sizes - 250 (left), 150 (middle) and 100 (right). Not including history does not effect the state values for larger LSTM but results in lower average value of the states observed for relatively smaller LSTM size.

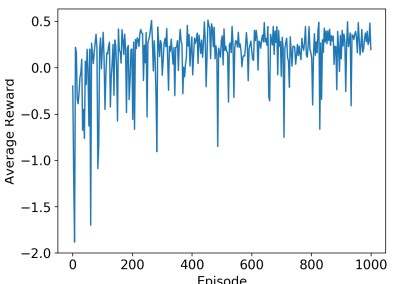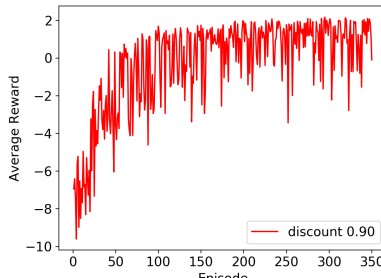

Figure 6: Plot of average reward observed in validation episodes with Q-agent (left) with $\gamma = 0.70$ and $\epsilon = 0.90$) and A3C agent (right) with $\gamma = 0.90$ and LSTM size $= 250$. The average reward value at convergence is larger for A3C agent than Q-agent.

marketplace. We collected feedback, from 12 designers, for both the search modalities across the different set of tasks. We asked the designers to rate our conversational search system on following metrics. Table 5 shows average rating value of each of these metrics.

1. **Information flow** to measure the extent to which the agent provide new information and suggestions which helped in driving the search forward (on a scale of 1 to 5 where 5 represents high information flow).

2. **Appropriateness** of actions to measure the suitability of actions taken by the agent during the search in terms of coherence (on a scale of 1 to 5 where 5 represents high appropriateness and denotes that it took right actions at right time during the search).

3. **Repetitiveness** to measure how repetitive was the agents actions in providing assistance during their search (on a scale of 1-5 where 1 represents not repetitive at all).

Table 5: Human Evaluation Ratings for Agent Trained Through A3C

| Metric | Average Rating |
|---|---|
| Information Flow | 2.58 |
| Appropriateness | 2.67 |
| Repetitiveness | 2.50 |

In addition to the above metrics, we also asked the designers to compare our system to conventional search interface of the stock photography marketplace in terms of following metrics :

1. **Engagement** : This is to measure how interactive and engaging conversational search is compared to conventional search on scale of 1 to 5 where 5 represents that conversational search is much more engaging and 1 represents same engagement as conventional search. We averaged the rating provided by the designers. Our system could achieve an average rating of 2.67 in this metric.

2. **Time Required**: We asked the designers to compare the two search modalities in terms of time required to search and reach to desired search results. We asked them to choose one of three options - conversational search required, 1.more time, 2. About the same time, 3. Less time (faster), than conventional search. About 33.3% of designers said that it requires more time while 16.7% said that conversational search reduced the time required to search the desired results. The remaining 50% believed that it required about the same time.

3. **Ease of using conversational search compared to conventional search** : We asked them to choose one of three options - conversational search is, 1. Difficult to use and adds additional burden, 2. About the same to use, 3. Much easier to use, compared to conventional search. 33.3% of the designers believed that conversational search is easier than conventional search, 41.7% said that it is same as conventional search while 25% believed that it is difficult to perform conversational search than conventional search.

The above evaluation shows that although we trained the bootstrapped agent through user model, it performs decently well with actual users by driving their search forward with appropriate actions without being much repetitive. The comparison with the conventional search shows that conversational search is more engaging. In terms of search time, it resulted in more search time for some designers while it reduces overall time required to search the desired results in some cases, in majority cases it required about the same time. The designers are regular users of conventional search interface and well versed with it, even then majority of them did not face any cognitive load while using our system with 33.3% of them believing that it is easier than conventional search.

## 5 CONCLUSION

In this paper, we develop a Reinforcement Learning based search assistant to interact with customers to help them search digital assets suited to their use-case. We model the rewards, state space, action space and develop an A3C based architecture which leverages the context of search to predict the policy. The trained agent is able to obtain higher average rewards in the validation episodes with virtual user and observes states with better values indicative of providing better search experience. We also propose a virtual stochastic user model to interact and train the RL agent in absence of labeled conversational data which accelerates the process of obtaining a bootstrapped agent.

As the next step, we would deploy our system to collect true conversational data which can be used to fine tune the current model as well as to train a new model which can generate the natural language responses in addition to deciding the action. In different search domains, designing the state and action space can take significant time which makes every situation an absolutely new task to be solved. To approach this issue as a future work, another system can be designed which helps in the automation of state space characterization with the help of system query logs.

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

# 6 APPENDIX

## 6.1 IMPLEMENTATION DETAILS

In this section, we discuss the multi-component architecture developed to facilitate conversational search. It consists of chat interface, Natural Language Unit (NLU), search engine and a module for RL agent. Figure 7 illustrates various architectural components and flow of control between different modules during one dialogue turn in a conversation.

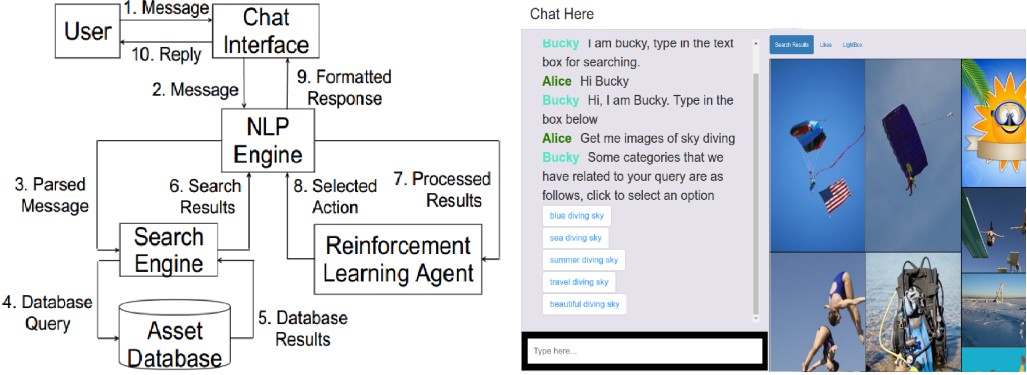

Figure 7: Architecture Diagram(left) and Chat-Search Interface(right)

### 6.1.1 CHAT INTERFACE

Chat interface comprises of a two-pane window, one for text dialogues and other for viewing search results. The chat interface allows the user to convey queries in form of dialogue in an unrestricted format. The user message is parsed by NLU which deduces user action from the input message. The NLU additionally obtains and redirects the query to search engine if the user action is $new\ query$ or $refine\ query$. User action and search engine output are forwarded to RL agent by NLU. The RL agent performs an action according to the learned policy based on which a formatted response is displayed on chat interface by the NLU. In addition to inputting the message, the chat interface provides functionality for other user actions such as liking a search result, adding assets to cart etc.

### 6.1.2 NATURAL LANGUAGE UNIT

The NLU is a rule based unit which utilizes dependencies between words and their POS tags in a sentence to obtain query and user action.[3] The query is formulated sequentially by locating the primary keyword and then determining the surrounding keywords. A distinction between the user actions $new\ query$ and $refine\ query$ is made depending on whether the current primary keyword is present in previous queries. A database is used to store the user queries to incorporate context of conversation while formulating the search query to be sent to the search engine. Similarly rules and stop words are used for determining different user actions. For instance, words like 'yes/'no' are used for providing feedback signals when the agent asks different probing questions, words like 'more' means that the user is requesting more results for the same query, 'cluster category click' determined based on whether the user clicks on one of the categorical options provided by the agent, 'search similar' is determined on the basis of whether the user drags the url of one of the search results into the text area.

### 6.1.3 SEARCH ENGINE

The search engine seeks assets from asset database corresponding to the search query formulated by NLU. The asset database comprises of about 5 million assets of different types like images, videos etc. Assets are indexed in asset database using tags and other metadata for retrieval during the search.[4]. The search engine also fetches and clusters assets having tags similar to most relevant assets retrieved to formulate related categorical asset options. Relevance scores of retrieved assets are calculated on the basis of the degree of match between asset metadata and the query. These scores are dispatched with fetched assets and categorical options to the NLU.

## 6.2 SOME CONVERSATIONS BETWEEN TRAINED AGENT AND REAL USERS

**Example 1**

**User** : hello
**Agent** : Hello, how may I help you?
**User** : i want images of cars
**Agent** : Categories might help you get better responses, click on the options below - sporty cars, expensive cars, city cars, sedan cars
**User** : city cars
**Agent** : Here are some of the images
**User** : show more
**Agent** : Do you want to browse through the following options? - urban city cars, wrecked city cars, traffic city cars
**User** : urban city cars
**Agent** : Your cart is the place where you can add the images you like. Click on the add to cart icon
**User** : <User adds few assets to cart>
**Agent** : Could you refine your query further so I can get you better images?
**User** : racing
**Agent** : Your cart is the place where you can add the images you like. Click on the add to cart icon

---

[3]User message is parsed using Stanford CoreNLP v3.6.0 toolkit available at `http://stanfordnlp.github.io/CoreNLP/other-languages.html`

[4]Assets are indexed using Apache Solr, a text-based search platform

**User** : <User drags an asset into text area to search for similar ones>
**Agent** : What are you going to use the images for?
**User** : i am organizing a racing competition
**Agent** : Here you go, these are some of the best matches for your query
**User** : <User adds few assets to cart>
**Agent** : Would you like to add something to your query?
**User** : no, bye

**Example 2**

**Agent** : Hi, type in the box below
**User** : show some cpu
**Agent** : Here are some of the images
**User** : can you show some more
**Agent** : Check out some images that we have
**User** : <User drags an asset into text area to search for similar ones>
**Agent** : Where will you use these images?
**User** : in my mac
**Agent** : Results for your query
**User** : <User adds few assets to cart>
**Agent** : I can sign you up, search images for you, add them to your cart and much more. Type in the box to chat
**User** : show me more
**Agent** : Is there anything else you want to search?
**User** : yes
**Agent** : We also have many images in these categories - Computer golden cpu, Technology golden cpu, Internet golden cpu, Abstract golden cpu, Chip golden cpu
**User** : internet golden cpu
**Agent** : Want me to get you anything else?
**User** : no, thanks for the help.

**Example 3**

**Agent** : You may enter your search query
**User** : show me some windows
**Agent** : Check out the images
**User** : more
**Agent** : Here are some of the images
**User** : more
**Agent** : Check out some images that we have
**User** : <User search for assets similar to one of the search results displayed>
**Agent** : Refine your query further to get better results
**User** : reflecting windows
**Agent** : Would you like to add something to your collections now? You can simply click on the add to cart icon
**User** : <User adds few assets to cart>
**Agent** : Could you refine your query further so I can get you better images?
**User** : shining reflecting windows
**Agent** : You could add the images to your cart for better exploration, click on the cart icon
**User** : no
**Agent** : Where will you use these images?
**User** : i will use them for construction designs
**Agent** : Here you go, these are some of the best matches for your query

**Example 4**

**User** : show some images of mountains
**Agent** : We also have many images in these categories, should I show you some?

**User** : nature mountains
**Agent** : Where will you use these images?
**User** : i am organizing adventure sports
**Agent** : Check out the images
**User** : can you show some more
**Agent** : Images can be added to cart for building a separate collection of stuff that you like. You can simply click on add to cart button
**User** : <User adds few assets to cart>
**Agent** : Want me to get you anything else?
**User** : i want people riding bike on mountains
**Agent** : We also have many images in these categories, should I show you some? - high mountains, hiking sports mountains, mountaineering mountains
**User** : hiking sports mountains
**Agent** : Signing up takes a moment, just give me your email id
**User** : ABC@XYZ.com

