# OpenReview forum: "Improving Search Through A3C Reinforcement Learning Based Conversational Agent"
_ICLR.cc/2018/Conference — Reject_

### Official Review · AnonReviewer1 · 2017-11-26
**Lack of context**

**Rating:** 2
**Confidence:** 5

**Review:**

This paper proposes to use RL (Q-learning and A3C) to optimize the interaction strategy of a search assistant. The method is trained against a simulated user to bootstrap the learning process. The algorithm is tested on some search base of assets such as images or videos.

My first concern is about the proposed reward function which is composed of different terms. These are very engineered and cannot easily transfer to other tasks. Then the different algorithms are assessed according to their performance w.r.t. to these rewards. They of course improve with training since this is the purpose of RL to optimize these numbers. Assessment of a dialogue system should be done according to metrics obtained through actual interactions with users, not according to auxiliary tasks etc.

But above all, this paper incredibly lacks of context in both RL and dialogue systems. The authors cite a 2014 paper when it comes to refer to Q-learning (Q-learning was first published in 1989 by Watkins). The first time dialogue has been casted into a RL problem is in 1997 by E. Levin and R. Pieraccini (although it has been suggested before by M. Walker). User simulation has been proposed at the same time and further developed in the early 2000 by Schatzmann, Young, Pietquin etc. Using LSTMs to build user models has been proposed in 2016 (Interspeech) by El Asri et al. Buiding efficient reward functions for RL-based conversational systems has also been studied for more than 20 years with early work by M. Walker on PARADISE (@ACL 1997) but also via inverse RL by Chandramohan et al (2011). A2C (which is a single-agent version of A3C) has been used by Strub et al (@ IJCAI 2017) to optimize visually grounded dialogue systems. RL-based recommender systems have also been studied before (e.g. Shani in JMLR 2005).

I think the authors should first read the state of the art in the domain before they suggest new solutions.

---

> ### Author Response · Authors · 2018-01-05
> **Reward Function and Evaluation**
>
> Thanks for your reviews.
>
> We have modeled rewards specifically for the domain of digital assets search in order to obtain a bootstrapped agent which performs reasonably well in assisting humans in their search so that it can be fine tuned further based on interaction with humans. As our problem caters to a subjective task of searching digital assets which is different from more common objective tasks such as reservation, it is difficult to determine generic rewards based on whether the agent has been able to provide exact information to the user unlike objective search tasks where rewards are measured based on required information has been provided to the user. This makes rewards transferability between subjective and objective search difficult. Though our modeled rewards are easily transferable to search tasks such as e-commerce sites where search tasks comprises of a subjective component (in addition to objective preferences such as price).
>
> Since we aim to optimise dialogue strategy and do not generate dialogue utterances, we assign the rewards corresponding to the appropriateness of the action performed by the agent considering the state and history of the search. We have used some rewards such as task success (based on implicit and explicit feedback from the user during the search) which is also used in PARADISE framework [1]. At the same time several metrics used by PARADISE cannot be used for modelling rewards. For instance, time required (number of turns) for user to search desired results cannot be penalised since it can be possible that user is finding the system engaging and helpful in refining the results better which may increase number of turns in the search.
>
> We evaluated our system through humans and added the results to the paper, please refer to section 4.3 in the updated paper. You may refer to appendix (section 6.2) for some conversations between actual users and the trained agent.
>
> Thanks for suggesting related references, we have updated our paper based on the suggestions. Kindly suggest any other further improvements.
>
> [1] Walker, Marilyn A., et al. "PARADISE: A framework for evaluating spoken dialogue agents." Proceedings of the eighth conference on European chapter of the Association for Computational Linguistics. Association for Computational Linguistics, 1997.

---

### Official Review · AnonReviewer2 · 2017-11-27
**lack of details**

**Rating:** 3
**Confidence:** 5

**Review:**

The paper describes reinforcement learning techniques for digital asset search.  The RL techniques consist of A3C and DQN.  This is an application paper since the techniques described already exist.  Unfortunately, there is a lack of detail throughout the paper and therefore it is not possible for someone to reproduce the results if desired.  Since there is no corpus of message response pairs to train the model, the paper trains a simulator from logs to emulate user behaviours.  Unfortunately, there is no description of the algorithm used to obtain the simulator.  The paper explains that the simulator is obtained from log data, but this is not sufficient.  The RL problem is described at a very high level in the sense that abstract states and actions are listed, but there is no explanation about how those abstract states are recognized from the raw text and there is no explanation about how the actions are turned into text.  There seems to be some confusion in the notion of state.  After describing the abstract states, it is explained that actions are selected based on a history of states.  This suggests that the abstract states are really abstract observations.   In fact, this becomes obvious when the paper introduces the RNN where a hidden belief is computed by combining the observations.  The rewards are also described at a hiogh level, but it is not clear how exactly they are computed.  The digital search application is interesting, however a detailed description with comprehensive experiments are needed for the publication of an application paper.

---

> ### Author Response · Authors · 2018-01-05
> **Details of User Model**
>
> Due to legal issues, we cannot not share the query session logs data. We have tried to provide details of our algorithm which can be used for obtaining user model from any given session logs data. The mapping between interactions in session log data and user actions which the agent can understand has been discussed in table 3. Using these mapping, we obtain a probabilistic user model (algorithm has been described in section 3.5). Figure 1 in the paper demonstrates how interactions in a session can be mapped to user actions.
>
> Kindly mention the sections which are lacking details and missing information in the algorithm for user model which will help us in improving our paper.

---

> ### Author Response · Authors · 2018-01-05
> **State and Reward Modeling**
>
> Thanks for your reviews.
>
> Our state representation comprises of history of actions taken by the user and the agent (along with other variables as described in the state space section 3.3) and not only the most recent action taken by the user. User action is obtained from user utterance using a rule-based Natural language unit (NLU) which uses dependency tree based syntactic parsing, stop words and pre-defined rules (as described in appendix, section 6.1.2). We capture the search context by including the history of actions taken by the user and the agent in the state representation. The state at a turn in the conversation comprises of agent and user actions in last ‘k’ turns. Since a search episode can extend indefinitely and suitability & dependence of action taken by the agent can go beyond last ‘k’ turns, we include an LSTM in our model which aggregates the local context represented in state (‘local’ in terms of state including only the recent user and agent actions) into a global context to capture such long term dependencies. We analyse the trend in reward and state values obtained by comparing it with the case when we do not include the history of actions is state and let the LSTM learn the context alone (section 4.1.3).
>
> Our system does not generate utterances, it instead selects an utterance based on the action taken by the agent from a corpus of possible utterances. This is because we train our agent to assist user in their search through optimising dialogue strategy and not actual dialogue utterances made by the agent. Though we aim to pursue this as future work where we generate agent utterances and train NLU for obtaining user action in addition to optimising dialogue strategy (which we have done in our current work).
>
> Since we aim to optimise dialogue strategy and do not generate dialogue utterances, we assign the rewards corresponding to the appropriateness of the action performed by the agent considering the state and history of the search. We have used some rewards such as task success, extrinsic rewards based on feedback signals from the user and auxiliary rewards based on performance on auxiliary tasks. These rewards have been modelled numerically on a relative scale.
>
> We have evaluated our model through humans and updated the paper, please refer to section 4.3 for human evaluation results and appendix (section 6.2) for conversations between actual users and trained agent.

---

### Official Review · AnonReviewer3 · 2017-11-27
**An interesting problem but a not convincing experimental protocol**

**Rating:** 5
**Confidence:** 4

**Review:**

The paper "IMPROVING SEARCH THROUGH A3C REINFORCEMENT LEARNING BASED CONVERSATIONAL AGENT" proposes to define an agent to guide users in information retrieval tasks. By proposing refinements of the query, categorizations of the results or some other bookmarking actions, the agent is supposed to help the user in achieving his search. The proposed agent is learned via reinforcement learning.

My concern with this paper is about the experiments that are only based on simulated agents, as it is the case for learning. While it can be questionable for learning (but we understand why it is difficult to overcome), it is very problematic for the experiments to not have anything that demonstrates the usability of the approach in a real-world scenario. I have serious doubts about the performances of such an artificially learned approach for achieving real-world search tasks. Also, for me the experimental section is not sufficiently detailed, which lead to not reproducible results. Moreover, authors should have considered baselines (only the two proposed agents are compared which is clearly not sufficient).

Also, both models have some issues from my point of view. First, the Q-learning methods looks very complex: how could we expect to get an accurate model with 10^7 states ? No generalization about the situations is done here, examples of trajectories have to be collected for each individual considered state, which looks very huge (especially if we think about the number of possible trajectories in such an MDP). The second model is able to generalize from similar situations thanks to the neural architecture that is proposed. However, I have some concerns about it: why keeping the history of actions in the inputs since it is captured by the LSTM cell ? It is a redondant information that might disturb the process. Secondly, the proposed loss looks very heuristic for me, it is difficult to understand what is really optimized here. Particularly, the loss entropy function looks strange to me. Is it classical ? Are there some references of such a method to maintain some exploration ability. I understand the need of exploration, but including it in the loss function reduces the interpretability of the objective (wouldn't it be preferable to use a more classical loss but with an epsilon greedy policy?).


Other remarks:
   - In the begining of "varying memory capacity" section, what is "100, 150 and 250" ? Time steps ? What is the unit ? Seconds ?
   - I did not understand the "Capturing seach context at local and global level" at all
   - In the loss entropy formula, the two negation signs could be removed

---

> ### Author Response · Authors · 2018-01-05
> **Experimental Details**
>
> We evaluated our system through real humans and added the results in section 4.3. Please refer to appendix (section 6.2) for some conversations between actual users and trained agent. For performing experiments with humans, we developed chat interface where an actual user can interact with the agent during their search. The implementation details of the chat interface have been discussed in the appendix (section 6.1.1). User action is obtained from user utterance using a rule-based Natural language unit (NLU) which uses dependency tree based syntactic parsing, stop words and pre-defined rules (as described in appendix, section 6.1.2). You may refer to supplementary material (footnote-2, page-9) which contains a video demonstrating search on our conversational search interface.
>
> In order to evaluate our system with the virtual user, we simulate validation episodes between the agent and the virtual user after every training episode. This simulation comprises of sequence of alternate actions between the user and the agent. The user action is sampled using the user model while the agent action is sampled using the policy learned till that point. Corresponding to a single validation episode, we determine two performance metrics. First is total reward obtained at the end of the episode. The values of the states observed in the episode is obtained using the model, average of states values observed during the validation episode is determined and used as the second performance metric. Average of these values over different validation episodes is taken and depicted in figures 3,4,5 and 6.

---

> ### Author Response · Authors · 2018-01-05
> **Q-learning and A3C System Modeling**
>
> Q-Learning Model:
> We experimented with Q-learning approach in order to obtain baseline results for the task defined in the paper since RL has not been applied before for providing assistance in searching digital assets. The large size of the state space requires large amount training data for model to learn useful representations since number of parameters is directly proportional to the size of state space which is indicative of the complexity of the model. The number of training episodes is not a problem in our case since we leverage the user model to sample interactions between the learning agent and user. This indeed is reflected in figure 6 (left), which shows that the model converges when trained on sufficient number of episodes.
>
> Since our state space is discrete, we have used table storage method for Q-learning. Kindly elaborate on what does generalisation of state means in this context so that we may elaborate more and improve our paper.
>
>
> A3C Model:
>
> We capture the search context by including history of actions taken by the user and the agent in last ‘k’ turns explicitly in the state representation. Since a search episode can extend indefinitely and suitability & dependence of action taken by the agent can go beyond last ‘k’ turns, we include an LSTM in our model which aggregates the local context represented in state (‘local’ in terms of including only the recent user and agent actions) to capture such long term dependencies and analyse the trend in reward and state values obtained by comparing it with the case when we do not include the history of actions in the state and let the LSTM learn the context alone (section 4.1.3).
>
> In varying memory capacity, by LSTM size (100,150,250), we mean dimension of the hidden state h of the LSTM. With more number of units, the LSTM can capture much richer latent representations and long term dependencies. We have explored the impact of varying the hidden state size in the experiments (section 4.1.2).
>
>
> Entropy loss function has been studied to provide exploration ability to the agent while optimising its action strategy in the Actor-Critic Model [1]. While epsilon-greedy policy has been successfully used in many RL algorithms for achieving exploration vs exploitation balance, it is commonly used in off-policy algorithms like Q-learning where the policy is not represented explicitly. The model is trained on observations which are sampled following epsilon-greedy policy which is different from the actual policy learned in terms of state-action value function.
>
> This is in contrast to A3C where we apply an on-policy algorithm such that the agent take actions according to the learned policy and is trained on observations which are obtained using the same policy. This policy is optimized to both maximise the expected reward in an episode as well as to incorporate the exploration behavior (which is enabled by using the exploration loss). Using epsilon-greedy policy will disturb the on-policy behavior of the learned agent since it will then learn on observations and actions sampled according to epsilon-greedy policy which will be different from the actual policy learnt which we represent as explicit output of our A3C model.
>
> The loss described in the paper optimise the policy to maximise the expected reward obtained in an episode where the expectation is taken with respect to different possible trajectories that can be sampled in an episode. In A3C algorithm, the standard policy gradient methods is modified by replacing the reward term by an advantage term which is difference between reward obtained by taking an action and value of the state which is used as a baseline (complete derivation in [2]). The learned baseline enforces that parameters are updated in a way that likelihood of actions that results in rewards better than value of the state is increased while it is decreased for those which provide rewards lower than the average action in that state.
>
>
>
> [1] : Mnih, Volodymyr, et al. "Asynchronous methods for deep reinforcement learning." International Conference on Machine Learning. 2016.
> [2] : Sutton, R. et al., Policy Gradient Methods for Reinforcement Learning with Function Approximation, NIPS, 1999)

---

> ### Author Response · Authors · 2018-01-05
> **A3C and rollouts are better than REINFORCE**
>
> Thanks for your reviews.
>
> Standard REINFORCE method for policy gradient has high variance in gradient estimates [1]. Moreover while optimising and weighing the likelihood for performing an action in a given state, it does not measure the reward with respect to a baseline reward due to which the agent is not able to compare different actions. This may result in gradient pointing in wrong direction since it does not know how good an action is with respect to other good actions in a given state. This may weaken the probability with which the agent takes the best action (or better actions).
>
> It has been shown that if a baseline value for a state is used to critic the rewards obtained for performing different actions in that state reduces the variance in gradient estimates as well as provides correct appraisal for an action taken in a given state (good actions get a positive appraisal) without requiring to sample other actions [2]. Moreover it has been shown that if baseline value of the state is learned through function approximation, we get an an unbiased or very less biased gradient estimates with reduced variance achieving better bias-variance tradeoff. Due to these advantages we use A3C algorithm since it learns the state value function along with the policy and provides unbiased gradient estimator with reduces variance.
>
> In standard policy gradient methods, multiple episodes are sampled before updating the parameters using the gradients obtained over these episodes. It has been observed that sampling gradients over multiple episodes which can span over large number of turns results in higher variance in the gradient estimates due to which the model takes more time to learn [3]. The higher variance is the result of stochastic nature of policy since taking sampling random actions initially (when the agent has not learned much) over multiple episodes before updating the parameters compounds the variance. Due to this reason, we instead use truncated rollouts where we update parameters of the policy and value model after every n-steps in an episode which are proven to be much effective and results in faster learning.
>
> [1] : Sehnke, Frank, et al. "Parameter-exploring policy gradients." Neural Networks 23.4 (2010): 551-559.
> [2] : Sutton, Richard S., et al. "Policy gradient methods for reinforcement learning with function approximation." Advances in neural information processing systems. 2000
> [3] : Tesauro, Gerald, and Gregory R. Galperin. "On-line policy improvement using Monte-Carlo search." Advances in Neural Information Processing Systems. 1997. ;  Gabillon, Victor, et al. "Classification-based policy iteration with a critic." (2011).

---

### Author Response · Authors · 2018-01-05
**We evaluated our system by performing human evaluation and updated our paper with corresponding results, please refer to section 4.3 in the updated paper.**

We evaluated our system trained using A3C algorithm through professional designers who regularly use image search site for their design tasks and asked them to compare our system with conventional search interface in terms of engagement, time required and ease of performing the search. In addition to this we asked them to rate our system on the basis of information flow, appropriateness and repetitiveness. The evaluation shows that although we trained the bootstrapped agent through user model, it performs decently well with actual users by driving their search forward with appropriate actions without being much repetitive. The comparison with the conventional search shows that conversational search is more engaging. In terms of search time, it resulted in more search time for some designers while it reduces the time required to search the desired results in some cases, in majority cases it required about the same time. The designers are regular users of conventional search interface and well versed with it, even then majority of them did not face any cognitive load while using our system with one-third of them believing that it is easier than conventional search.

---

### Decision · Program_Chairs · 2018-01-29
**ICLR 2018 Conference Acceptance Decision**

**Decision:**

Reject

**Comment:**

meta score: 4

This paper is primarily an application paper applying known RL techniques to dialogue.    Very little reference to the extensive literature in this area.

Pros:
 - interesting application (digital search)
 - revised version contains subjective evaluation of experiments

Cons:
 - limited technical novelty
 - very weak links to the state-of-the-art, missing many key aspects of the research domain